# High-Risk Human Papillomaviruses and Epstein–Barr Virus in Colorectal Cancer and Their Association with Clinicopathological Status

**DOI:** 10.3390/pathogens9060452

**Published:** 2020-06-08

**Authors:** Ishita Gupta, Halema Al Farsi, Ayesha Jabeen, Faruk Skenderi, Hamda Al-Thawadi, Yaman M. AlAhmad, Ibrahim Abdelhafez, Ala-Eddin Al Moustafa, Semir Vranic

**Affiliations:** 1College of Medicine, QU Health, Qatar University, 2713 Doha, Qatar; ishugupta28@gmail.com (I.G.); halfarsi@qu.edu.qa (H.A.F.); jabeen@qu.edu.qa (A.J.); halthawadi@qu.edu.qa (H.A.-T.); ym1407016@qu.edu.qa (Y.M.A.); ia1409078@student.qu.edu.qa (I.A.); 2Department of Pathology, Clinical Center, University of Sarajevo, 71000 Sarajevo, Bosnia and Herzegovina; faruk.skenderi@gmail.com; 3Biomedical Research Centre, Qatar University, 2713 Doha, Qatar

**Keywords:** Epstein–Barr virus, human papillomaviruses, rectal cancer, immunohistochemistry, PCR

## Abstract

Colorectal cancer (CRC) is a common malignancy with a high mortality rate worldwide. It is a complex, multifactorial disease that is strongly impacted by both hereditary and environmental factors. The role of microbes (e.g., viruses) in the pathogenesis of CRC is poorly understood. In the current study, we explored the status of high-risk human papillomaviruses (HPV) and Epstein–Barr virus (EBV) in a well-defined CRC cohort using immunohistochemistry and polymerase chain reaction assays. Our data showed that high-risk HPVs were common (~80%) and EBV had a low presence (14–25%) in the CRC samples. The most common high-risk HPVs are HPV16, 31, 18, 51, 52 and 45 genotypes. The co-presence of high-risk HPV and EBV was observed in ~16% of the sample population without any significant association with the clinicopathological variables. We conclude that high-risk HPVs are very prevalent in CRC samples while EBV positivity is relatively low. The co-expression of the two viruses was observed in a minority of cases and without any correlation with the studied parameters. Further studies are necessary to confirm the clinical relevance and potential therapeutic (preventive) effects of the observations reported herein.

## 1. Introduction

Colorectal cancer (CRC) is the third most common cancer reported worldwide, and corresponds to about 10% of all cancer cases [1]. It is considered to be second most common cause of mortality in both developed and developing countries, affecting men and women equally [2].

CRC is a complex disease involving multiple factors including hereditary (family history and genetic diseases such as familial adenomatous polyposis (FAP) and hereditary non-polyposis CRC (Lynch) syndromes), (epi)genetic, lifestyle, environmental and viral factors [3].

Among viral factors, microbial–epithelial interactions are assumed to instigate oncogenic activity, which could lead to the onset and development of CRC [3]. Viral infections by high-risk human papillomaviruses (HPVs) and Epstein–Barr virus (EBV) have been linked with the initiation and progression of several human carcinomas [4,5,6].

HPV is a small epitheliotropic, non-enveloped, double-stranded DNA virus that can be sexually transmitted. It infects the epidermal or mucosal epithelial cells, where HPVs can induce neoplastic transformation (both benign and malignant) [7,8]. The multiplication of the virus occurs in cell nuclei and is firmly linked to cell differentiation [9]. The HPV genome has eight open reading frames (ORFs) and is divided into three functional elements (the early (E), and late (L)regions, and a long control region (LCR)) [10]. The E) region is vital for processes like control of viral transcription as well as replication and it encodes six proteins (E1, E2, E4, E5, E6 and E7) [11]. The L region takes part in virus assembly by encoding the structural proteins (L1-L2) [10], whereas the high-risk HPV early proteins (E5, E6 and E7) act as oncogenes, working closely together through molecular mechanisms involved in the epithelial–mesenchymal transition (EMT) [11,12]. These proteins deregulate cell cycle progression, immortalize epithelial cells, which leads to their neoplastic transformation in cooperation with other oncogenes and/or oncoviruses [8,13,14]. Several studies have confirmed the etiological role of HPVs in cervical [15] and head and neck carcinomas [16] while the role of HPVs in breast [17] and bladder carcinomas [18] has been also proposed and reported.

On the other hand, EBV is a double-stranded DNA gamma-herpes virus, which infects 90% of the adult population [2,19]. EBV causes infectious mononucleosis and various epithelial and lymphoproliferative malignancies such as nasopharyngeal and gastric carcinoma as well as several B-cell lymphomas and T-cell/NK lymphomas [19]. EBV genome is 184-kbp long and encodes over 85 genes including viral oncogenes such as six EBV-encoded nuclear antigens (EBNA1, -2, -3A, -3B, -3C and -LP) and latent membrane proteins (LMP1, -2A, and -2B), as well as various noncoding RNAs (EBERs and miRNAs) [2,20]. LMP1, oncogenic protein of EBV, is known to stimulate cell growth, reduce apoptosis, promote cell motility, and angiogenesis [21,22]. During EBV infection, EBNA1 is the only protein articulated in all forms of latency [23]. EBNA1 is considered as an oncoprotein and links EBV infection to carcinogenesis [24].

Although previous studies have indicated the etiological role of HPV in rectal cancer [25,26,27], the underlying mechanisms are still nascent. The role of EBV in rectal cancer is even less understood [28]. In the current study, we explored the co-presence of high-risk HPVs and EBV in a well-defined rectal cancer cohort.

## 2. Results

### 2.1. Clinicopathological Characteristics of the Cohort

The clinicopathological characteristics of the cohort are summarized in Table 1. The study included samples from 64 (60%) male and 42 (40%) female patients. The mean age of the patients was 65 years (range, 41–86 years) and the vast majority of specimens were taken from the rectum (102 cases, 96%). The remaining four samples were from the sigmoid colon and other parts of the large bowel (Table 1).

All tumors were histologically adenocarcinomas. The majority of cancers were grade two (moderately differentiated) (65 cases, 62%) (Figure 1A,B) while 39 cases (37%) were grade three (poorly differentiated).

Only one case was well-differentiated (grade 1) rectal adenocarcinoma (Table 1). Notably, twenty-one cases (20%) had been previously treated with neoadjuvant chemotherapy or combined chemo- and radiotherapy. None of the cases had received targeted therapies (e.g., anti-EGFR, anti-angiogenic or immune checkpoint inhibitors) prior to the surgery.

### 2.2. The Status of High-Risk HPV Subtypes and EBV by PCR

The PCR data for HPVs and EBV and their association with the clinicopathological parameters are summarized in Table 2. 

The most commonly found high-risk HPVs in the entire cohort (n = 106) were HPV16 (53%) followed by HPV31 (51%), HPV18 (50%), HPV51 (46%), HPV52 (39%), HPV45 (39%), HPV35 (26%), HPV56 (9%) and HPV39 (0.9%).

Four common high-risk HPVs (16, 18, 31 and 35) were found in 82% of the cases, while twenty-five cases (22.6%) were co-infected with both HPV16 and 18. Notably, HPV16-/HPV18+ phenotype was significantly higher (75%) in males compared with HPV16+/HPV18- phenotype (48%) (*p* = 0.035).

On the other hand, 15/106 patients (14.1%) were positive for *EBNA1* alone and 26/106 (24.5%) were positive for the *LMP1* gene of EBV; 15/106 (14.1%) of the cases were positive for both (*EBNA1* and *LMP1*) sequences. Although all the EBNA1-positive tumors were LMP1-positive tumors, the same did not hold for LMP1-positive tumors. Only 16% (17/106) of the colorectal cases were positive for both high-risk HPVs and EBV.

Among the rectal cancer cases (n = 102), the four most common high-risk HPVs (16, 18, 31 and 35) were present in 83/102 (81%) of the cases, while twenty-three cases (22.5%) were co-infected with both HPV16 and 18 (Figure 2).

Moreover, the rectal cancers were co-infected with more than one type of HPV in 74 out of 102 samples (72.5%). Figure 3 shows the most common co-infections of rectal cancer samples with HPV16 and other high-risk HPV subtypes. Three or more co-infections were seen in 58/102 rectal cancer samples (57%), fourteen cases (13.6%) had three HPV co-infections, 18 cases (17.5%) had four co-infections, 11 cases (10.7%) had five co-infections, 12 cases (11.7%) had six co-infections while three cases (2.9%) had seven high-risk HPV co-infections (Appendix A). The most commonly expressed combinations were HPV16/18/51 (11/102, 10.8%) followed by HPV16/18/31 in 10/102 (9.8%) cases.

*EBNA1* and *LMP1* were individually present in 15/102 (14%) and 25/102 (24%) of the cases, respectively; 15/102 (14%) of the cases were positive for both (*EBNA1* and *LMP1*) sequences. Although all the EBNA1-positive tumors were LMP1-positive tumors, the same did not hold for LMP1-positive tumors.

To calculate the number of EBV cases, EBNA1 and LMP1 positive samples were taken together, which came to a total of 26 cases. The co-presence of EBV (EBNA1 and LMP1) and high-risk HPVs was detected in 16% (16/102) of the rectal cases (Table 3). In addition, a significant correlation between EBV and various HPV types (HPV16 (*p* = 0.02), HPV18 (*p* = 0.03), HPV35 (*p* = 0.03), HPV52 (*p* = 0.03) and HPV56 (*p* = 0.003) was found in the rectal cancer samples (χ^2^ test with Yates correction) (Table 3).

### 2.3. Immunohistochemical Expression of E6 (HPVs) and LMP1 (EBV)

The immunohistochemical (IHC) data for HPV and EBV and their relationship to the clinicopathological parameters are summarized in Table 4.

Interpretable results by IHC were obtained for 58 cases of HPVs. Forty-five cases (77.5%) were positive for E6 above the threshold of 1% of positive cancer cells; in contrast, LMP1 of EBV expression was found in 11% of the cases (7/63) (Table 4). Four out of seven LMP1-positive cases also co-expressed E6 of HPV (Figure 1C–F). When positive, both E6 of HPV and LMP1 of EBV expression was diffused (>50% cancer cells were positive) in the majority of the cases. Notably, adjacent normal mucosa was devoid of E6 while LMP1 expression was occasionally seen in tumor-infiltrating lymphocytes (TIL) and lymphatic aggregates within the tumor and adjacent structures.

Positive samples for E6 of HPV showed a statistically significant association with cancers of rectal origin (*p* = 0.024). In addition, LMP1 of EBV positive samples by IHC seemed to be associated with grade 2 adenocarcinomas (*p* = 0.035) (Table 4).

PCR and IHC data were in good concordance for HPV analysis while discrepant data were observed in EBV assessment; however, the inter-reliability rating between PCR and IHC was fair (Kappa = 0.31; *p* = 0.005).

## 3. Discussion

There are various types of HPV-associated cancers including cervical, vulvar, vaginal, penile, rectal, anal and oropharyngeal cancer [29]. From a clinical and therapeutic point of view, exploring HPV status in cancers could be highly relevant due to the prophylactic vaccines that have been shown to be effective in preventing common HPV-associated cancers such as cervical cancer [29].

In this investigation we explored for the first time, the co-presence of high-risk HPVs and EBV in human CRC in the Bosnian population; our study found a high prevalence of high-risk HPVs and a low positivity of EBV in CRC samples. The most frequent HPV types in our cohort are 16, 31, 18, 51, 52, and 45. This is also, to the best of our knowledge, the first report regarding the distribution of high-risk HPVs in CRC samples from Bosnia. Our data are similar to those reported for a large cohort of cervical cancer patients by de Sanjose et al. [30] and from Bosnian cervical cancer patients [31,32,33]. Among our HPV-positive rectal cases, HPV16 and HPV18 were the most prevalent genotypes (~50%), which is in line with the HPV distribution in cervical cancer samples that were previously reported in the Bosnian population [31,32,33]. In Polish CRC samples, HPV16 and 18 were also the most common HPV serotypes (~60%) [34]. Another study done on Italian CRC patients revealed HPV in 33% of the cases [35]. A meta-analysis study based on the European population showed that HPV18 was present in 47% of CRC cases [36]. The overall HPV16+/18+ prevalence in our study was ~23%, which is similar to the findings in cervical cancer samples from the Croatian population (21%) [37]. HPVs 16, 31 and 18 were also the most common high-risk HPVs in cervical cancer samples in Serbian women [38]. In our study, the other expressed HPV-types included HPV−45, −51 and −52, which have been previously reported in cervical cancer in Bosnia [33]. Moreover, HPV45 has been previously found in colorectal cancer [25,39]. Studies done in the USA and Iran identified HPV-51 as the most frequent HPV subtype in colorectal cancer [39,40].

The highest reported EBV-positivity in CRC was up to 46%, with the majority of the studies reporting EBV positivity of 20–40% [19,28,41,42]. Our study indicated the individual presence of EBNA1 and LMP1 genes of EBV in 14% and 25% of the samples, respectively. EBV consists of two strains, B95–8 (EBV type 1) and AG876 (EBV type 2), which also known as type A and type B, respectively [43]. These EBV variants are differentiated based upon genetic differences in EBNA sequences [43], and discrepancies in EBV variants have previously been described in breast cancer samples [44]. This can be explained by the different roles of *EBNA* and *LMP* genes. In general, EBNA1 acts as a transcriptional activator that binds to the FR element, thereby enhancing the expression of the viral Cp promoter and *LMP* genes. This in turn leads to transcriptional activation of viral *LMP* genes (LMP-1 and -2) promoting cellular proliferation [45,46,47]. During latency 0, no EBNA protein is expressed, while only one EBNA (EBNA1) is expressed during latency I and II, and during latency III, all six EBNAs are expressed [48,49,50]. Further, the co-presence of both EBNA1 and LMP1 in our cohort can be explained by the fact that in EBV-infected cells, in the latent phase, EBNA1 and LMP1 are expressed in a cancer-specific manner [51]. Thus, the most restricted pattern of viral latent gene expression (latency 1: EBNA1 and EBERs) is expressed in Burkitt lymphomas [51,52] while the unrestricted pattern of viral gene expression (latency II and III: EBNA1, LMPs and EBERs) is expressed in Hodgkin lymphomas, diffused large B cell lymphomas and EBV-positive gastric and nasopharyngeal carcinomas [51,52]. Regardless of the latency pattern, virtually all EBV-associated malignancies express EBNA-1 and EBERs [52]. In addition, the functional variance can be explained by the genetic diversity of the *LMP-1* gene, which shows a higher degree of polymorphisms than most EBV genes [53,54]. Ten main LMP1 variants (China 1, China 2, China 3, NC (North Carolina), Mediterranean+ (Med+), Med-, GD1, SEA 1, and SEA 2) have been defined in different regions and diseases [55,56,57].

Karpinski et al. reported that 19% of the CRC cases from Poland were EBV positive [58]. In an Italian CRC cohort, the EBV-positivity rate ranged from 3–29% according to RT-PCR and sequencing assays [59]. Another study in Italy did not find any significant involvement of any oncogenic virus (HPV, HTLV, HHV-8, JCV, BKV and Merkel cell polyomavirus), except for EBV (52%) [60]. However, a Czechoslovakian study failed to detect any oncogenic virus (EBV, HPV and CMV) in adenocarcinoma of the colon [61]. A Chinese cohort of colorectal carcinomas was reported to be EBV positive (LMP1) in 27% of the samples [62].

The combined oncogenic effects of viral infections have been recognized as potential oncogenic drivers in various cancers [6]. Oncoviruses can lead to the onset and progression of cancer via commonly shared pathways including WNT/β-catenin, JAK/STAT/SRC, PI3k/Akt/mTOR, and/or RAS/MEK/ERK signaling pathways [6,8,14,41]. A study conducted by Guidry and Scott [63] showed that co-infection by HPV and EBV enhances EBV persistence either via latency or increased viral replication or by aggregating HPV oncogene expression. Our group has previously demonstrated that oncoproteins of EBV and high-risk HPVs can interact during the onset of human carcinomas, and subsequently result in metastasis progression via epithelial–mesenchymal transition [14]. Epithelial differentiation stimulates both the productive and lytic phases of HPV and EBV, respectively; high-risk HPV stabilizes the EBV genome and induces EBV lytic reactivation in differentiating epithelial cells, which suggests that co-infection with HPV may increase EBV-mediated pathogenesis of CRC [64]. Additionally, HPV and EBV may work together to promote the proliferation of cultured cervical [65] and prostate cells [66], suggesting the same for colorectal epithelial cells. In our study, co-expression of EBV and HPV was seen in only ~16% of CRCs. This co-expression was not associated with any noted clinicopathological parameters. The lack of any association may be due to the relatively small sample size as well as low EBV positivity in the cohort.

## 4. Materials and Methods

### 4.1. Sample Collection and DNA Extraction

The study included 106 formalin-fixed paraffin-embedded (FFPE) tissue samples that were diagnosed in 2010–2017 at the Department of Pathology, University Clinical Center Sarajevo, Bosnia and Herzegovina. For the study, all cases were de-identified and patients’ information anonymized. The study was conducted based on the IBC approval of the local research committee (#IBC-2019/005).

Prior to molecular assays, all cases were re-reviewed by board-certified pathologists (F.S. and S.V.) to confirm the diagnosis and select appropriate FFPEs for tissue microarray, immunohistochemistry and PCR assays.

DNA was extracted from FFPE tissue samples using the Thermo Scientific GeneJET FFPE DNA Purification Kit according to the manufacturer’s instructions (ThermoFisher Scientific, USA). Briefly, FFPE sections were subjected to enzymatic digestion using 200 µL of digestion buffer. After digestion, 20 µL Proteinase K solution was added to lyse and release genomic DNA, which was de-crosslinked by heat incubation (90 °C) for 40 min. The obtained solution was further centrifuged and the supernatant containing DNA was mixed with 200 µL binding buffer. After addition of ethanol (96%), the lysate was loaded onto the purification column and the adsorbed DNA was subjected to washing (wash Buffers 1 and 2) to eliminate contaminants. DNA was eluted with 60µL elution buffer.

### 4.2. EBV and HPV Detection by PCR

Twenty-five nanograms of purified genomic DNA from each sample was analyzed for HPV and EBV by polymerase chain reaction (PCR) as previously described [67] using specific primers for HPV types: 16, 18, 31, 33, 35, 39, 45, 51, 52, 56 and 58 as well as *EBNA1* (Forward Primer: 5′-GAGCGGGGAGATAATGTACA-3′ and Reverse Primer: 5′-TAAAAGATGGCCGGACAAGG-3′) and *LMP1* (Forward Primer: 5′-TTGGAGATTCTCTGGCGACT-3′ and Reverse Primer: 5′-AGTCATCGTGGTGGTGTTCA-3′) of EBV. GAPDH primers (Forward Primer: 5′-GAAGGC-CATGCCAGTGAGCT-3′ and Reverse Primer: 5′ CCGGGAAACTGTGGCGTGAT-3′) were used as internal control. All the primers and analyses were performed as previously described [68].

PCR was performed using the Invitrogen Platinum II Hot-Start Green PCR Master Mix (2X) (ThermoFisher Scientific, Waltham, MA, USA). Briefly, *HPV* gene was amplified for an initial denaturation at 94 °C for 2 min followed by 40 cycles of 94 °C for 30 s, annealing at temperatures ranging from 50 to 62 °C for 30 s depending on each primer’s melting temperature as previously described [68], and 72 °C for 30 s. In parallel, *EBNA1* and *LMP1* genes were amplified for an initial denaturation at 94 °C for 2 min followed by 40 cycles of 94 °C for 30 s, 61 °C for 30 s, and 72 °C for 30 s. The samples were finally incubated for 10 min at 72 °C for a final extension. The PCR product from each exon was resolved using 1.5% agarose gel electrophoresis.

### 4.3. Tissue Microarray (TMA)

Tissue microarray was constructed as described previously by our group [17]. Briefly, cancer samples and controls were embedded into a virgin paraffin TMA block using a manual tissue arrayer (Beecher Instruments, Silver Spring, MD, USA).

Following the sampling of two TMA cores of 1.0 mm in diameter, sections of 4 µm were cut and stained with hematoxylin and eosin (H&E) on the initial slides to verify the histopathological diagnosis. Next, slides of completed blocks were used for immunohistochemical assays.

### 4.4. Immunohistochemistry (IHC)

Immunohistochemical assays for E6 (HPV) and LMP1 (EBV) expression were performed using procedures as previously described [17]. For TMA construction, each slide was deparaffinized in graded alcohol, rehydrated and boiled (microwave) in 10 mM citrate sodium solution (pH 6.0) for 10 min. Endogenous peroxidase activity within the rehydrated tissue was blocked with a solution of 3% hydrogen peroxide in methanol for 10 min at room temperature. TMA slides were further incubated for 35 min at 37 °C with primary monoclonal antibodies for E6 (clones 1–4 and C1P5, Dako Agilent, Carpinteria, CA, USA and Calbiochem, Canada) and LMP1 (clone CS1–4, Abcam) using a fully automated immunostainer (Ventana Medical System, Tucson, AZ). The fully automated Ventana Medical System uses an indirect biotin–avidin system with a universal biotinylated immunoglobulin secondary antibody. The slides were counterstained with hematoxylin prior to mounting. The staining procedures were completed according to the manufacturer’s recommendations. Negative controls were obtained by omitting specific primary antibodies for E6 and LMP1 as well as specific blocking peptides from Santa Cruz Biotechnology.

The tumors were considered positive for E6 and LMP1 if cancer cells exhibited positivity ≥1% of the cells [69]. In the case of LMP1 protein expression, we also evaluated the presence of viral infection in tumor-infiltrating lymphocytes and stromal cells [69,70].

### 4.5. Statistical Analysis

The Spearman Correlation Rank test was used to assess the significance of HPV and EBV associations. The Kolmogorov–Smirnov test and histograms were used to assess normal distribution. Categorical variables were compared using Chi square and Kruskal–Wallis tests while Cramer’s V test was used to assess a possible association between nominal and ordinal variables in asymmetric tables. HPV and EBV IHC status were included as the dependent variables in a multiple regression analysis to test the relationship with other independent variables such as age, gender, tumor location, stage, and grade. The values for missing samples were excluded from the regression analysis. Cohen’s kappa coefficient was used to assess inter-rater reliability for categorical variables. The results were considered statistically significant if *p*-values were ≤0.05 in two-tailed tests. All the statistical analyses were performed using IBM Statistical Package for the Social Sciences (version 25).

## 5. Conclusions

We conclude that high-risk HPVs are prevalent in CRC samples while EBV positivity is relatively low. The co-presence of the two viruses was observed in a minority of the cases and without any correlation with the studied parameters. Further studies are necessary to confirm the clinical and preventive relevance of the observed findings.

## Figures and Tables

**Figure 1 pathogens-09-00452-f001:**
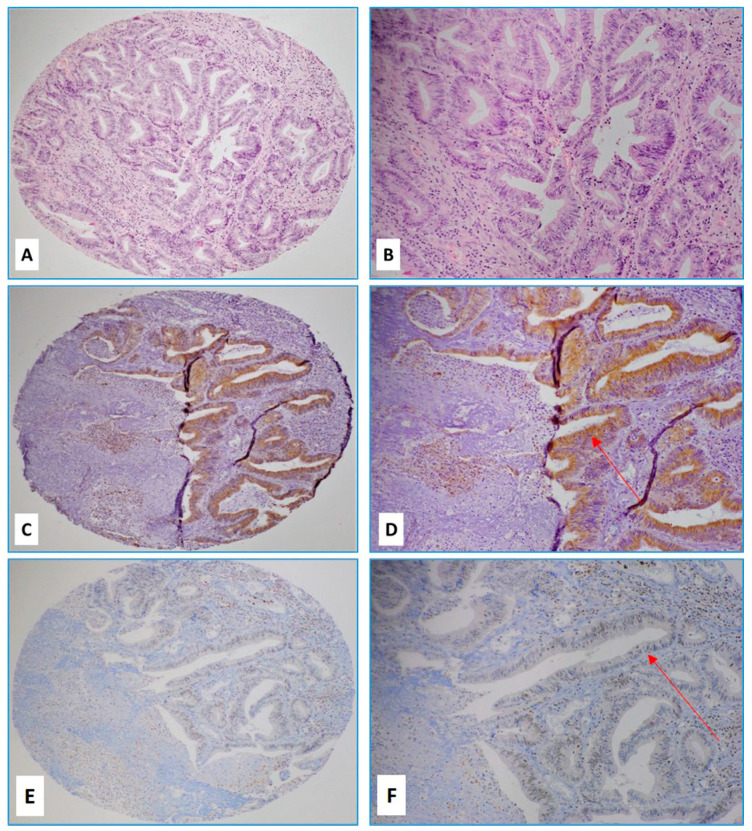
(**A**,**B**) Hematoxylin and eosin (H&E) slides of a case of moderately differentiated rectal carcinoma with a strong (3+) intensity of E6 protein by immunohistochemistry (**C**,**D**) and moderate (2+) expression of LMP1 protein (**E**,**F**) (Red arrows indicate positive cancer cells; (**A**,**C**,**E**) images 4×, (**B**,**D**,**F**) images, 10× magnification).

**Figure 2 pathogens-09-00452-f002:**
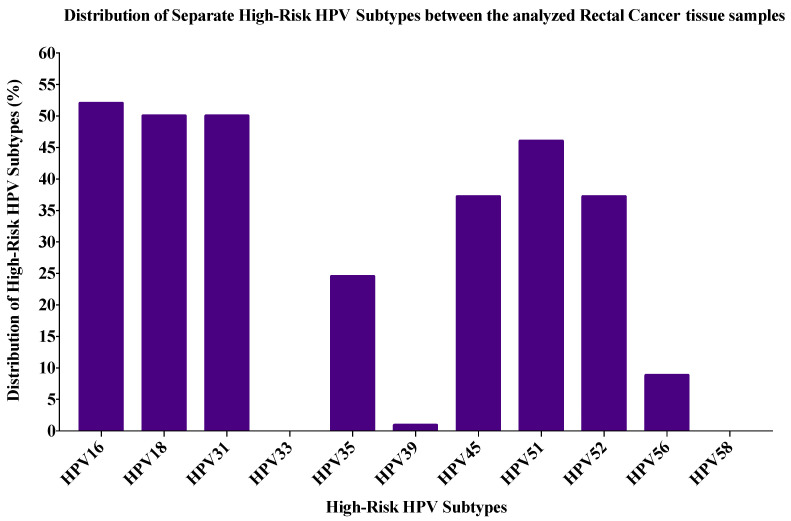
The distribution of each high-risk HPV subtype by PCR according to the frequency in the rectal cancer (RC) cohort (n = 102).

**Figure 3 pathogens-09-00452-f003:**
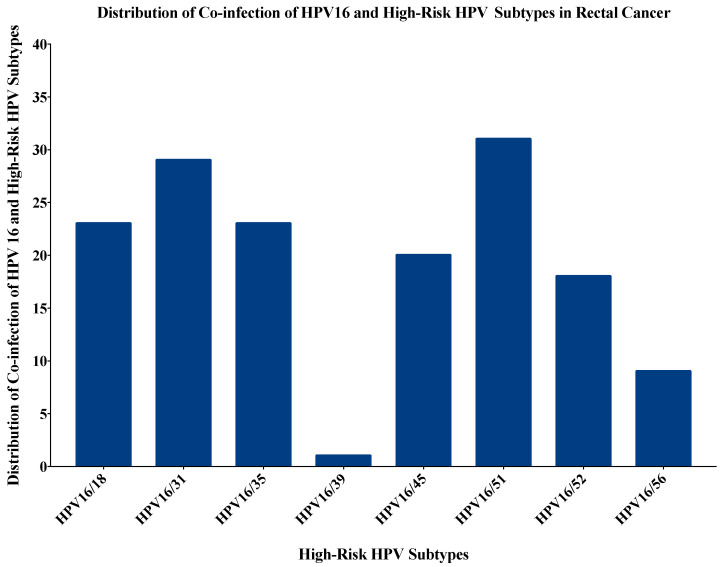
The distribution of co-infection of HPV16 with other high-risk HPV subtypes in rectal cancer. The most commonly co-infected subtype was HPV16/51 (31/102; 30.4%) followed by HPV16/31 (29/102; 28.4%), HPV16/18 and HPV16/35 (23/102; 22.5% each). Since, HPV33 and HPV58 were not present in the cohort, there was no co-expression for HPV16 with these subtypes.

**Table 1 pathogens-09-00452-t001:** The clinicopathological characteristics of the cohort.

Variable	Total (%) ^†^ n = 106
**Age (mean ± SD) 65 ± 8 yrs**	
**Tumor grade**	
Grade 1	1 (1)
Grade 2	65 (62)
Grade 3	39 (37)
**Tumor stage ***	
Stage I	22 (21)
Stage II	23 (22)
Stage III	54 (52)
Stage IV	5 (5)
**Sex**	
Male	64 (60)
Female	42 (40)
**Tumor origin**	
Rectum	102 (96)
Sigmoid colon	2 (2)
Other parts of colon	2 (2)

^†^ Numbers may not add up to the total number of subjects because of some missing values. * The staging was according to the American Joint Committee on Cancer (AJCC) TNM system (8th edition). SD = Standard deviation.

**Table 2 pathogens-09-00452-t002:** The relationship between human papillomaviruses (HPV) and Epstein–Barr virus (EBV) status by PCR and clinicopathological variables.

Variable	HPV (PCR)		EBV (PCR)	
	(−) n = 8 (%)	(+) n = 98 (%)	(−) n = 80 (%)	(+) n = 26 (%)
**Tumor grade**				
**Grade 1**	0 (0)	1 (1)	1 (1)	0 (0)
**Grade 2**	6 (75)	59 (61)	48 (61)	17 (65)
**Grade 3**	2 (25)	37 (38)	30 (38)	9 (35)
***p*-value**	**0.72**		**0.80**	
**Tumor stage ***				
**Stage I**	1 (12)	20 (21)	15 (19)	6 (23)
**Stage II**	2 (25)	22 (23)	19 (24)	5 (19)
**Stage III**	4 (50)	49 (51)	40 (51)	13 (50)
**Stage IV**	1 (12)	5 (5)	4 (5)	2 (8)
***p*-value**	**0.81**		**0.83**	
**Sex**				
**Male**	4 (50)	60 (61)	46 (58)	18 (69)
**Female**	4 (50)	38 (39)	34 (42)	8 (31)
***p*-value**	**0.53**		**0.29**	
**Tumor origin**				
**Rectum**	0 (0)	94 (96)	77 (96)	25 (96)
**Sigmoid**	0 (0)	2 (2)	2 (3)	0 (0)
**Other parts of colon**	0 (0)	2 (2)	1 (1)	1 (4)
***p*-value**	**0.84**		**0.51**	
**HPV (+)**				25 (96)
***p*-value**			**0.41**	

* The staging was according to the American Joint Committee on Cancer (AJCC) TNM system (8th edition). PCR = polymerase chain reaction; HPV = human papillomavirus; EBV = Epstein–Barr virus.

**Table 3 pathogens-09-00452-t003:** The prevalence of high-risk HPV types and their relationship to EBV status in the rectal cancer cohort (n = 102). Significant *p*-values are labeled with an asterisk.

Samples	No. of Cases	High-Risk HPV Types										
		16	18	31	33	35	39	45	51	52	56	58
**EBV (+)**	26	18	17	15	0	11	0	7	16	7	6	0
**EBV (−)**	76	30	30	37	0	14	0	30	30	6	2	0
**Total**	**102**	**48**	**47**	**52**	**0**	**25**	**0**	**37**	**46**	**13**	**8**	**0**
***p*-value**		***0.02 ****	***0.03 ****	***0.57***	***N/A***	***0.03 ****	***N/A***	***0.36***	***0.08***	***0.03 ****	***0.003 *****	***N/A***

**Table 4 pathogens-09-00452-t004:** The relationship between HPV and EBV expressions by immunohistochemical (IHC) and clinicopathological variables.

Variable	HPV (IHC)		EBV (IHC)	
	(−) n = 13 (%)	(+) n = 45 (%)	(−) n = 56 (%)	(+) n = 7 (%)
**Age (years)**				
***p*-value**	**0.44**		**0.76**	
**Tumor grade**				
**Grade 1**	0 (0)	0 (0)	0 (0)	0 (0)
**Grade 2**	8 (62)	27 (61)	33 (60)	7(100)
**Grade 3**	5 (38)	17 (39)	22 (40)	0 (0)
***p*-value**	**0.92**		**0.035**	
**Tumor stage ***				
**Stage I**	3 (25)	7 (16)	10 (18)	0 (0)
**Stage II**	1 (8)	6 (14)	8 (15)	1 (14)
**Stage III**	6 (50)	28 (64)	32 (58)	6 (86)
**Stage IV**	2 (17)	3 (6)	5 (9)	0 (0)
***p*-value**	**0.31**		**0.38**	
**Sex**				
**Male**	6 (50)	30 (68)	34 (62)	4 (57)
**Female**	6 (50)	14 (32)	21 (38)	3 (43)
***p*-value**	**0.32**		**0.78**	
**Tumor origin**				
**Rectum**	10 (83)	42 (95)	52 (95)	6 (86)
**Sigmoid**	0 (0)	2 (5)	2 (3)	0 (0)
**Other parts of colon**	2 (17)	0 (0)	1 (2)	1 (14)
***p*-value**	**0.024**		**0.19**	
**HPV (+)**				4 (57)
***p*-value**			**0.89**	

* The staging was according to the American Joint Committee on Cancer (AJCC) TNM system (8th edition). IHC = immunohistochemistry; HPV = human papillomavirus; EBV = Epstein–Barr virus.

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
