# Peer review of "High-Risk Human Papillomaviruses and Epstein–Barr Virus in Colorectal Cancer and Their Association with Clinicopathological Status"

_pathogens, 2020, doi:10.3390/pathogens9060452_

Round 1
Reviewer 1 Report
This manuscript describes the analysis of colorectal tumors for the presence of HPVs (several were tested) or EBV, using both PCR and immunohistochemistry to assess the presence of virus in Bosnian patients. The authors find that most rectal cancers are HPV+, while a minority are EBV+, and even fewer are positive for both viruses. There is no correlation between positivity for both viruses and clinico-pathological parameters, though this conclusion is weakened somewhat by the relatively small number of dual positives. The manuscript is well written, the results are present well, and the conclusions should be of interest to many readers.
The most significant issue involves a P value in Table 2. According to a chi-squared test, the difference in tumor origin among HPV- and HPV+ cancers is highly significant, P <10-18, not 0.84. I obtain P = 0.84 when I enter the "8" HPV- cancers as rectal tumors instead of sigmoid tumors in the table for the chi-squared test. Does the "8" belong in the "Rectal" row and was it mis-entered in Table 2 (which would be more consistent with Table 3), or does the "8" belong in the "Sigmoid" row as shown and was it mis-entered during statistical calculations? Either way, the error needs to be corrected in the table and/or the text.
Additional comments:
Please provide a figure showing EBV EBNA1 and LMP1 immunohistochemistry (analogous to Figure 1 showing HPV E6). Also, please report whether all EBNA1+ tumors are also LMP1+. If not, please discuss. Along these lines, the discussion of the excess of LMP1+ tumors relative to EBNA1+ tumors (Lines 166-183) is confusing. Please clarify.
Tables 2 and 3. Since the numbers may not add up to the total number of subjects because of missing values, please provide a number for the number of EBV- tumors that were HPV+ (second to last line of each table).
Line 19: "review" should be replaced with "manuscript" or equivalent.
Lines 46: A closing parenthesis is needed after "syndromes."
Line 51: HPV can be transmitted in non-sexual ways. Recommend replacing "is sexually transmitted" with "can be sexually transmitted."
Line 52: "mucosal epithelial cells in the skin" does not make sense. Recommend deleting "in the skin."
Line 57: The vast majority of HPV's do not encode E3 (which is why an earlier statement, about HPV's 8 open reading frames, is correct).
Line 70: "other" should be deleted from "other latent proteins."
Line 76: "While" should be deleted.
Line 186: Please explain how the results reported in the Italian and Czechoslovakian studies were "opposite" to those described in the previous sentences.
Line 243: Recommend "Tissue microarray was constructed as described . . ."
Author Response
This manuscript describes the analysis of colorectal tumors for the presence of HPVs (several were tested) or EBV, using both PCR and immunohistochemistry to assess the presence of virus in Bosnian patients. The authors find that most rectal cancers are HPV+, while a minority are EBV+, and even fewer are positive for both viruses. There is no correlation between positivity for both viruses and clinico-pathological parameters, though this conclusion is weakened somewhat by the relatively small number of dual positives. The manuscript is well written, the results are present well, and the conclusions should be of interest to many readers.
The most significant issue involves a P value in Table 2. According to a chi-squared test, the difference in tumor origin among HPV- and HPV+ cancers is highly significant, P <10-18, not 0.84. I obtain P = 0.84 when I enter the "8" HPV- cancers as rectal tumors instead of sigmoid tumors in the table for the chi-squared test. Does the "8" belong in the "Rectal" row and was it mis-entered in Table 2 (which would be more consistent with Table 3), or does the "8" belong in the "Sigmoid" row as shown and was it mis-entered during statistical calculations? Either way, the error needs to be corrected in the table and/or the text.
RESPONSE: Sorry, this was a typo error; the study had no sigmoid tumors that were negative for HPV by PCR. This is now corrected in Table 2 (page 5, line 124).
Additional comments:
Please provide a figure showing EBV EBNA1 and LMP1 immunohistochemistry (analogous to Figure 1 showing HPV E6). Also, please report whether all EBNA1+ tumors are also LMP1+. If not, please discuss. Along these lines, the discussion of the excess of LMP1+ tumors relative to EBNA1+ tumors (Lines 166-183) is confusing. Please clarify.
RESPONSE: We are not able to provide the image with EBNA1 IHC as we did not perform IHC for EBNA1. IHC assay for EBV was limited to LMP1 protein. However, PCR assay included probes for both LMP1 and EBNA1. It has been clarified in the results about EBNA1+ tumors expressing LMP1 as well (pages 5-6, lines 134-146; page 7, lines 162-165). In the discussion paragraph, we further explored the relationship between the EBNA1 and LMP1 expressions (page 9, line 239-240). Of note, the figure 1 was replaced by another figure following in part your and reviewer#2 request to highlight the co-presence of both HPV and EBV in colorectal cancer samples (page 4, lines 107-109).
Tables 2 and 3. Since the numbers may not add up to the total number of subjects because of missing values, please provide a number for the number of EBV- tumors that were HPV+ (second to last line of each table).
RESPONSE: We created a separate table (Table 3) to clarify the relationship between EBV positive and negative tumors and different HPV types (page 7, lines 170-171).
Line 19: "review" should be replaced with "manuscript" or equivalent.
RESPONSE: The word “review” has been replaced with “manuscript” as indicated (page 1, line 19).
Lines 46: A closing parenthesis is needed after "syndromes."
RESPONSE: The closing parenthesis has been added (page 2, line 49).
Line 51: HPV can be transmitted in non-sexual ways. Recommend replacing "is sexually transmitted" with "can be sexually transmitted."
RESPONSE: The suggested change has been made (page 2, line 54).
Line 52: "mucosal epithelial cells in the skin" does not make sense. Recommend deleting "in the skin."
RESPONSE: The phrase “in the skin” has been deleted as suggested (page 2, line 55).
Line 57: The vast majority of HPV's do not encode E3 (which is why an earlier statement, about HPV's 8 open reading frames, is correct).
RESPONSE: As suggested by the reviewer, E3 has been deleted (page 2, line 60).
Line 70: "other" should be deleted from "other latent proteins."
RESPONSE: The word “other” has been deleted as suggested (page 2, line 73).
Line 76: "While" should be deleted.
RESPONSE: The word “while” has been deleted as mentioned (page 2, line 79).
Line 186: Please explain how the results reported in the Italian and Czechoslovakian studies were "opposite" to those described in the previous sentences.
RESPONSE: The results about the Italian and Czechoslovakian studies have been included (page 10, lines 257-260).
Line 243: Recommend "Tissue microarray was constructed as described . . ."
RESPONSE: The suggested change has been made (page 11, line 319).
Reviewer 2 Report
Recently, more and more publications have appeared in scientific literature dedicated to human papillomaviruses (HPV) and Epstein–Barr virus (EBV) studies in oncology. This work is devoted to the study of high-risk HPV and EBV in colorectal cancer and their association with clinic-pathological status. Despite the fact that the article might be interesting to both clinicians and virologists who study the importance of viruses in carcinogenicity, the article contains a number of uncertainties/errors.
Major Comments:
Line 112 - In the Fig 2 the distribution is not very well perceived: 1) The distribution of the Y-axis should be more detailed, say every five; 2) The Y-axis notation should, however, include %, as it is not possible to understand whether these are cases or percentages, as the distribution can be represented in both ways; 3) the total percentage is 313.9%, it can be confusing, which means that there are dual (trial) infections and then these cases should be presented additionally, separately, because it is currently difficult to understand which combinations predominate and what their significance would be.
Lines 113-114 – If there are 25 cases co-infected with both HPV16 and 18 it should be shown also in the Figure 2;
Lines 116-117 - What means “EBNA1 and LMP1 genes of EBV were presented in 15 (14.1%) and 26 (24.4%) of the samples, respectively”? In the Table 2 as EBV PCR-positive are mentioned 26 samples in total. Does that mean that 15/26 had been positive for both sequences? It should be more clearly described.
Line 117 - For better understanding also the 10 (9.4%) cases positive for both high-risk HPVs and EBV should be shown in the PCR Table 2;
Line 119 - What does it mean “EBNA1 and LMP1 was 81%, 14% and 24% respectively”?
Line 119-120 - There is discrepancy in percentage of co-presence of HPVs and sequences: 13% there but 9.4% cases in the Line 117. Needs to be explained;
Lines 120-122 - “…there was a significant correlation of EBV with various HPV types…” – should be shown in numbers, not only by “p” value (could be in small Table);
Line 126 - The mean of the empty line in the Table “Age (years)” is not understandable as well as the penultimate line (HPV+) - should be explained.
Lines 135-136 - “When positive, both E6 of HPV and LMP1 of EBV expression was diffuse in majority of cases (Figure 1C-D)” doesn't suit the truth because in the Fig.1 only expression of E6 of HPV is shown (according to the legend); It would be desirable that this Fig. would show both HPV E6 and EBV LMP1 expression;
Lines 166-167 – “Our study indicates EBNA1 and LMP1 expression in 14% and 25% of samples” – is not correct, because it seems authors are speaking about PCR results not about antigen expression. In that case should be “Our study indicates presence of EBNA1 and LMP1 genes of EBV …”
Lines 167 – it is not clear what authors understand under “EBV variants” – needs to be explained.
Minor comments:
Line 75 - should be [23-25] instead of [23-26] because [26] is about EBV;
Line 91 - the percentage indication 65 cases (61%) does not match the indication in Table 1-65 (62%) (Line 86);
Line 109 - the number in cohort (106) does not much the number in cohort (102) in the legend of Figure 2;
Line 134 -should be Table 3 instead of Table 2;
Lines 152-153 –“…the most frequent HPV types in our cohort are 16, 31, 18, 51, 52 and 45…” but in the Abstract HPV16, 31, 18 and 51 – it would be better if it were the same, moreover, only the first 3 are discussed further in the Discussion.
I suggest adding two articles to the references; one of them is the work of the co-authors of this article, which would allow improving the discussion part:
- Fernandes, Q., Gupta, I., Vranic, S., Al Moustafa, A.E. Human Papillomaviruses and Epstein-Barr Virus Interactions in Colorectal Cancer: A Brief Review. 2020 Apr 20;9(4). pii: E300. doi: 10.3390/pathogens9040300
- Ferreira, A.R.; Ramalho, A.C.; Marques, M.; Ribeiro, D. The Interplay between Antiviral Signalling and Carcinogenesis in Human Papillomavirus Infections. Cancers (Basel). 2020 Mar 10;12(3). pii: E646. doi: 10.3390/cancers12030646.
Author Response
Recently, more and more publications have appeared in scientific literature dedicated to human papillomaviruses (HPV) and Epstein–Barr virus (EBV) studies in oncology. This work is devoted to the study of high-risk HPV and EBV in colorectal cancer and their association with clinic-pathological status. Despite the fact that the article might be interesting to both clinicians and virologists who study the importance of viruses in carcinogenicity, the article contains a number of uncertainties/errors.
Major Comments:
Line 112 - In the Fig 2 the distribution is not very well perceived: 1) The distribution of the Y-axis should be more detailed, say every five; 2) The Y-axis notation should, however, include %, as it is not possible to understand whether these are cases or percentages, as the distribution can be represented in both ways; 3) the total percentage is 313.9%, it can be confusing, which means that there are dual (trial) infections and then these cases should be presented additionally, separately, because it is currently difficult to understand which combinations predominate and what their significance would be.
RESPONSE: In figure 2 we have made the following changes (page 6, lines 151-152): 1) The distribution of the Y-axis has been modified to five as suggested; 2) The Y-axis notation now includes the percentage symbol (%) as suggested; 3) A new figure (Figure 3) has been added that indicates a combination of HPV 16 with the other HPV subtypes to explain the domination of each subtype (page 7, lines 160-161).
Lines 113-114 – If there are 25 cases co-infected with both HPV16 and 18 it should be shown also in the Figure 2;
RESPONSE: Cases co-infected with HPV16/18 have been included as the last bar in figure 2. The figure legend has been updated (page 6, lines 151-152).
Lines 116-117 - What means “EBNA1 and LMP1 genes of EBV were presented in 15 (14.1%) and 26 (24.4%) of the samples, respectively”? In the Table 2 as EBV PCR-positive are mentioned 26 samples in total. Does that mean that 15/26 had been positive for both sequences? It should be more clearly described.
RESPONSE: In the results section, it has been defined that 14.1% were positive for EBNA1 only and 24.5% were positive for LMP1 alone. It has been further added that 14.1% of the cases were positive for both, EBNA1 and LMP1 genes of EBV (pages 5-6, lines 134-146).
Line 117 - For better understanding also the 10 (9.4%) cases positive for both high-risk HPVs and EBV should be shown in the PCR Table 2;
RESPONSE: We created a separate table (table 3) to highlight the relationship between EBV and different HPV types by PCR (page 7, lines 170-171). We point out here that the 10% was based on the co-presence between HPV16/18 and EBNA1 of LMP1. However, if all the high-risk HPVs and EBV status are taken, the total percentage that reflects the co-presence is 16%. This is now corrected in the abstract (page 1, line 30) and in the results paragraph (page 6, line 147; page 7, line 167).
Line 119 - What does it mean “EBNA1 and LMP1 was 81%, 14% and 24% respectively”?
RESPONSE: The explanation for each of the percentages has been explained (page 6, line 150; page 7, lines 163-164).
Line 119-120 - There is discrepancy in percentage of co-presence of HPVs and sequences: 13% there but 9.4% cases in the Line 117. Needs to be explained;
RESPONSE: The discrepancy has been explained; 16% indicates co-presence of HPV and EBV in the total cohort of 106 colorectal cases (page 6, line 147), while 16% indicates co-presence of HPV and EBV in the 102 rectal cases (page 7, lines 167-168). We also clarified the relationship between EBNA1 and LMP1 in the entire and rectal cohort, respectively (pages 5-6, lines 135-151; page 7, lines 163-168).
Lines 120-122 - “…there was a significant correlation of EBV with various HPV types…” – should be shown in numbers, not only by “p” value (could be in small Table);
RESPONSE: Table 3 has been included in the results section showing the prevalence of high-risk HPV types in EBV(+) and EBV(-) rectal cancer cohort (102 cases) (page 7, line 171-172).
Line 126 - The mean of the empty line in the Table “Age (years)” is not understandable as well as the penultimate line (HPV+) - should be explained.
RESPONSE: This has been now corrected in Table 1 (page 3, line 95).
Lines 135-136 - “When positive, both E6 of HPV and LMP1 of EBV expression was diffuse in majority of cases (Figure 1C-D)” doesn't suit the truth because in the Fig.1 only expression of E6 of HPV is shown (according to the legend); It would be desirable that this Fig. would show both HPV E6 and EBV LMP1 expression;
RESPONSE: The figure has been replaced (page 4). The legend has also been added accordingly (page 4, lines 108-110).
Lines 166-167 – “Our study indicates EBNA1 and LMP1 expression in 14% and 25% of samples” – is not correct, because it seems authors are speaking about PCR results not about antigen expression. In that case should be “Our study indicates presence of EBNA1 and LMP1 genes of EBV …”
RESPONSE: The suggested change has been made (page 9, lines 230-231).
Lines 167 – it is not clear what authors understand under “EBV variants” – needs to be explained.
RESPONSE: EBV variants have been defined as suggested (page 9, lines 231-233).
Minor comments:
Line 75 - should be [23-25] instead of [23-26] because [26] is about EBV;
RESPONSE: The suggested change has been made (page 2, line 79).
Line 91 - the percentage indication 65 cases (61%) does not match the indication in Table 1-65 (62%) (Line 86).
RESPONSE: Apologies for the typing error. The suggested change has been made (page 3, line 101).
Line 109 - the number in cohort (106) does not much the number in cohort (102) in the legend of Figure 2;
RESPONSE: Figure 2 represents a distribution of high-risk HPV in rectal cancer; as mentioned in results it states that our total colorectal cancer cohort was 106; of these 106, we had 102 rectal cancer cases and 4 cases from other parts of the large bowel (Section 2, page 2, line 87).
Line 134 -should be Table 3 instead of Table 2.
RESPONSE: Due to the addition of Table 3 (page 7, line 171-172), the table number has now been changed to Table 4 and the suggested change has been made (page 8, line 192).
Lines 152-153 –“…the most frequent HPV types in our cohort are 16, 31, 18, 51, 52 and 45…” but in the Abstract HPV16, 31, 18 and 51 – it would be better if it were the same, moreover, only the first 3 are discussed further in the Discussion.
RESPONSE: The HPV types – 52 and -45 have now been included in the abstract (page 1, line 29). A few sentences have been added in the discussion to maintain consistency (page 9, lines 225-228).
I suggest adding two articles to the references; one of them is the work of the co-authors of this article, which would allow improving the discussion part:
Fernandes, Q., Gupta, I., Vranic, S., Al Moustafa, A.E. Human Papillomaviruses and Epstein-Barr Virus Interactions in Colorectal Cancer: A Brief Review. 2020 Apr 20;9(4). pii: E300. doi: 10.3390/pathogens9040300
Ferreira, A.R.; Ramalho, A.C.; Marques, M.; Ribeiro, D. The Interplay between Antiviral Signalling and Carcinogenesis in Human Papillomavirus Infections. Cancers (Basel). 2020 Mar 10;12(3). pii: E646. doi: 10.3390/cancers12030646.
RESPONSE: Both references have been added in the manuscript as Ref 6 and 8, respectively.
Round 2
Reviewer 2 Report
The paper is has been partially improved, but unfortunately there are uncorrected/ misunderstood parts that need to be corrected, as the article cannot be published in its current form.
- Lines 29-30 in the Abstract - The most common high-risk HPVs are HPV16, 31, 18, 51, 52 and 45 genotypes detected in >50% of the samples” do not correspond to the data in the Figure 2 and textual part in the Lines 108-110 (Results) – only : only 3 HPV genotypes (16, 18, 51) are detected in <50% of the samples. Needs to be corrected in the Abstract.
- The Fig 1 is supplemented, but at least in 10x magnification images (D and F), at least some positive cells should be indicated by arrows, as is usually done in IHC images.
- Fig.2 – in reality in this Fig. authors show distribution of separate High-Risk HPV Subtypes between the analysed Rectal Cancer tissue samples therefore it must also be reflected in the title “Distribution of separate High-Risk HPV Subtypes between the analysed Rectal Cancer tissue samples”.
- The Fig. 3 is completely not understandable and did not reflect “Distribution of Co-infection of HPV16 and High-Risk HPV Subtypes in Rectal Cancer”. Co-infection is the simultaneous infection of a host by multiple pathogen species. In virology, co-infection includes simultaneous infection of a single cell or tissue sample by two or more virus particles. Co-infection of HPV16 and other high-risk HPV subtypes is not shown in this Fig. and in general it reflects the same data as in the Fig. 2 only in the wrong way. Moreover, in Lines 119-121 authors are writing 23 cases (22.5%) were co-infected with both HPV16 and 18 and it is shown also in the Fig. 2 but it is not seen in the Fig.3. It is also not understandable why on the Y axis there are more than 100% of samples which is nonsense. What authors understand under “The graph also denotes co-presence of HPV16/18 in the rectal cancer cohort” (Line 123) – is it double infection (co-infection) which is not shown, moreover separate infections within a cohort already are shown in the Fig. 2.
- According to the data in the Fig. 2 there should be samples containing sequences not only of 2 HVP genotypes bet even more – maybe 3 etc. If the authors do not have such data the Fig. 3 should be deleted, if the authors have such data (better in real numbers but not as a percentage) the Fig. 3 should be properly designed.
- Lines 129-130 – is written “EBNA1 and LMP1 were individually present in 15/102 (14%) and 25/102 (24%) of the cases, respectively; 15/102 (14%) of the cases were positive for both (EBNA1 and LMP1) sequences” which means all EBNA1 positive cases were also LMP1 positive – all together 25 cases. In the Table 3 (Line 138) you are speaking about 26 EBV positive cases - where you get it and which number is correct? Please add to the Table which method did you use for analysis in this case.
- Lines 150-151 – authors are writing “Four out of seven LMP1-positive cases also co-expressed E6 of HPV (Figure 1C-F)” but it is not seen in reality. If there is co-expression it must be shown in the Fig. 1D-F and indicated by arrows.
Author Response
The paper has been partially improved, but unfortunately there are uncorrected/misunderstood parts that need to be corrected, as the article cannot be published in its current form.
Lines 29-30 in the Abstract - The most common high-risk HPVs are HPV16, 31, 18, 51, 52 and 45 genotypes detected in >50% of the samples” do not correspond to the data in the Figure 2 and textual part in the Lines 108-110 (Results) – only : only 3 HPV genotypes (16, 18, 51) are detected in <50% of the samples. Needs to be corrected in the Abstract.
Response: First, we would like to thank the reviewer for his/her constructive comments regarding our work. As requested, the % detected in the samples has been deleted from the abstract (page 1, line 30).
The Fig 1 is supplemented, but at least in 10x magnification images (D and F), at least some positive cells should be indicated by arrows, as is usually done in IHC images.
Response: Red arrows have now been added to highlight the positive cancer cells for E6 (HPV) and LMP1 (EBV) (page 4, line 94).
2 – in reality in this Fig. authors show distribution of separate High-Risk HPV Subtypes between the analysed Rectal Cancer tissue samples therefore it must also be reflected in the title “Distribution of separate High-Risk HPV Subtypes between the analysed Rectal Cancer tissue samples”.
Response: The title of Figure 2 has been changed as suggested by the reviewer (page 6, line 124).
The Fig. 3 is completely not understandable and did not reflect “Distribution of Co-infection of HPV16 and High-Risk HPV Subtypes in Rectal Cancer”. Co-infection is the simultaneous infection of a host by multiple pathogen species. In virology, co-infection includes simultaneous infection of a single cell or tissue sample by two or more virus particles. Co-infection of HPV16 and other high-risk HPV subtypes is not shown in this Fig. and in general it reflects the same data as in the Fig. 2 only in the wrong way. Moreover, in Lines 119-121 authors are writing 23 cases (22.5%) were co-infected with both HPV16 and 18 and it is shown also in the Fig. 2 but it is not seen in the Fig.3. It is also not understandable why on the Y axis there are more than 100% of samples which is nonsense. What authors understand under “The graph also denotes co-presence of HPV16/18 in the rectal cancer cohort” (Line 123) – is it double infection (co-infection) which is not shown, moreover separate infections within a cohort already are shown in the Fig. 2.
Response: The legend for Figure 2 has been revised (page 6, lines 125-126). Figure 3 and its legend have also been modified as suggested by the reviewer (page 7, lines 132-136).
According to the data in the Fig. 2 there should be samples containing sequences not only of 2 HVP genotypes bet even more – maybe 3 etc. If the authors do not have such data the Fig. 3 should be deleted, if the authors have such data (better in real numbers but not as a percentage) the Fig. 3 should be properly designed.
Response: As requested, Figure 3 has been replaced by a new figure that reflects HPV16 co-infections with other high-risk HPVs (real numbers and not as a percentage, page 7, line 132). Given that the figure that would reflect all the cases with >2 co-infections was difficult to create, we updated the results paragraph with the number of the cases that had > 2 high-risk HPV co-infections and added a supplemental table 1 that contains all the possible co-infections including multiple high-risk HPV co-infections.
Lines 129-130 – is written “EBNA1 and LMP1 were individually present in 15/102 (14%) and 25/102 (24%) of the cases, respectively; 15/102 (14%) of the cases were positive for both (EBNA1 and LMP1) sequences” which means all EBNA1 positive cases were also LMP1 positive – all together 25 cases. In the Table 3 (Line 138) you are speaking about 26 EBV positive cases - where you get it and which number is correct? Please add to the Table which method did you use for analysis in this case.
Response: We are sorry about this miscalculation, to calculate the number of EBV cases, EBNA1 and LMP1 positive samples were taken into account, which gave the 26 cases total and hence has been included in the text (page 7, lines 141-142). The statistical analysis was done using χ2 test with Yates correction and has been mentioned in the text (page 7, line 142) as well as in the title of Table 3 (page 7, line 144).
Lines 150-151 – authors are writing “Four out of seven LMP1-positive cases also co-expressed E6 of HPV (Figure 1C-F)” but it is not seen in reality. If there is co-expression it must be shown in the Fig. 1D-F and indicated by arrows.
Response: The figure with IHC stains of E6 and LMP1 proteins was revised and red arrows are added to highlight the positive cancer cells for E6 (HPV) and LMP1 (EBV) (page 4, line 94).
Round 3
Reviewer 2 Report
Compared to the original version, the article has been significantly improved.